# A Comprehensive Mass Spectrometry-Based Workflow for Clinical Metabolomics Cohort Studies

**DOI:** 10.3390/metabo12121168

**Published:** 2022-11-24

**Authors:** Zhan Shi, Haohui Li, Wei Zhang, Youxiang Chen, Chunyan Zeng, Xiuhua Kang, Xinping Xu, Zhenkun Xia, Bei Qing, Yunchang Yuan, Guodong Song, Camila Caldana, Junyuan Hu, Lothar Willmitzer, Yan Li

**Affiliations:** 1Metanotitia Inc., No 59. Gaoxin South 9th Road, Yuehai Street, Nanshan District, Shenzhen 518056, China; 2The First Affiliated Hospital of Nanchang University, No.17 Yongwaizheng Street, Nanchang 330209, China; 3The Second Xiangya Hospital of Central South University, Furong District, Changsha 410011, China; 4The Second Hospital of Tianjin Medical University, No 23. Pingjiang Road, Hexi District, Tianjin 300211, China; 5Max Planck Institute of Molecular Plant Physiology, Potsdam Science Park, Am Muehlenberg 1, 14476 Potsdam, Germany

**Keywords:** metabolomics, clinical cohort, LC-MS, GC-MS, quality control, data normalization, data modeling

## Abstract

As a comprehensive analysis of all metabolites in a biological system, metabolomics is being widely applied in various clinical/health areas for disease prediction, diagnosis, and prognosis. However, challenges remain in dealing with the metabolomic complexity, massive data, metabolite identification, intra- and inter-individual variation, and reproducibility, which largely limit its widespread implementation. This study provided a comprehensive workflow for clinical metabolomics, including sample collection and preparation, mass spectrometry (MS) data acquisition, and data processing and analysis. Sample collection from multiple clinical sites was strictly carried out with standardized operation procedures (SOP). During data acquisition, three types of quality control (QC) samples were set for respective MS platforms (GC-MS, LC-MS polar, and LC-MS lipid) to assess the MS performance, facilitate metabolite identification, and eliminate contamination. Compounds annotation and identification were implemented with commercial software and in-house-developed *PAppLine*^TM^ and Ulib^MS^ library. The batch effects were removed using a deep learning model method (NormAE). Potential biomarkers identification was performed with tree-based modeling algorithms including random forest, AdaBoost, and XGBoost. The modeling performance was evaluated using the F1 score based on a 10-times repeated trial for each. Finally, a sub-cohort case study validated the reliability of the entire workflow.

## 1. Introduction

Metabolites are intermediates or end products of biosynthetic reactions that can directly depict the physiological status of cells [1,2]. In comparison to the other “omics” techniques, namely genomics, transcriptomics, and proteomics, metabolomics is closer to the phenotype in a biological system, enabling us to better understand the inner interaction between genetics, health status, and environmental impacts [3]. With the development of mass spectrometry (MS)-based technologies and machine learning approaches, metabolomics is becoming a promising tool for precision medicine [4,5]. In recent years, metabolomics has been successfully applied in several clinical studies, for example, in the diagnosis of the inborn errors of metabolism (IEM) [6] and the discovery of biomarkers for diagnosing cancers, Alzheimer’s diseases, cardiovascular disease, and obesity disease [7,8,9,10,11,12,13]. 

Metabolomics strategies have been divided into two distinct approaches for clinical studies: targeted and untargeted metabolomics. Targeted metabolomics is the measurement of a known set of metabolites. The number of samples in targeted clinical metabolomics studies is relatively small (normally < 200). This may lead to some bias, resulting in a high chance of false discovery [14]. In contrast, untargeted clinical metabolomics focuses on the comprehensive profiling of all the measurable metabolites in a large group of biological samples. Due to its comprehensive nature and the extensive amount of data generated, this approach faces challenges to achieve precise results. For example, the quality of clinical samples can be easily influenced by environmental factors (e.g., temperature, handling time). Compared with other cellular biomolecules, metabolite species and concentrations changed rapidly in vivo and even in vitro. Hence, well-practiced standard operation procedures (SOPs) for sample collection, storage, and preparation are vital for the success of metabolomics experiments [15]. Equivalently, a reliable instrument analysis method is required to warrant the quality of data acquisition. Metabolites represent a diverse class of molecules, including amino acids, peptides, lipids, nucleic acids, organic acids, carbohydrates, vitamins, and thiols. These compounds process the diversity of physicochemical properties (i.e., structure, size, polarity, and volatility), which leads to the impossibility of a global analysis through one analytical platform or single analysis method. Using multiple analytical platforms is a desirable strategy to expand the coverage and depth of metabolite mining. Untargeted clinical metabolomics analysis offers the opportunity for novel biomarker discovery. However, the principal challenges of this approach lie in the standardization of sample collection and preparation, quality control, batch effect correction, and data processing and analysis. The objective of this work was to develop a comprehensive guide for clinical metabolomics cohort studies from pre-acquisition (clinical sample harvest, sample quality inspection, transportation, and storage) and acquisition (sample preparation and metabolomics data acquisition) to post-acquisition (data curation and model selection and evaluation) (Figure 1). In this study, clinical metabolomics was based on plasma samples, which were collected from large-scale and multi-center cancer cohorts in China (ca. 2500 participants). Untargeted metabolomics profiling of these plasma samples was performed using three MS platforms, namely, untargeted liquid chromatography (LC)-MS polar and lipid and targeted GC-MS. The performance of our workflow was evaluated using a sub-cohort study, from which we identified not only well-reported biomarkers but also a group of novel candidates for further validation. These results demonstrated that our clinical metabolomics workflow study allows reproducible global metabolomics profiling of large clinical samples with a broad coverage and a high accuracy, which could pave the way for applications of big data-driven metabolomics in clinical studies. 

Clinical samples were collected as described in the SOP (Appendix A) and screened via crucial exclusion criteria. Before data acquisition, strict quality assurance was performed. Three types of QC samples were set for evaluation. “All-in-one extraction” was chosen to cover as many types of metabolites as possible by not introducing too many extraction steps and solvents. A combination of commercial and in-house-developed libraries and software was applied for metabolite annotation. Data analysis was performed as described in the methods. 

## 2. Material and Methods

### 2.1. Study Design and Biospecimen Choice 

A case-control study enrolling 2490 participants, belonging to three cancers groups, namely, lung cancer (LC), gastric cancer (GC), and colorectal cancer (CRC), as well as two control groups (healthy control (HCtrl) and diseased/non-cancer (NC)), was performed between September 2018 and January 2020. The subjects were registered in three Chinese hospitals: The First Affiliated Hospital of Nanchang University (NCH; Nanchang, China), the Second Xiangya Hospital of Central South University (XYH; Changsha, China), and the Second Hospital of Tianjin Medical University (TJH; Tianjin, China). The study was approved by their local Institutional Review Board, following the guidelines of the International Conference on Harmonization for Good Clinical Practice and the Declaration of Helsinki, including the formal consent of the participants. Prior to sampling, all participants were screened using the inclusion and exclusion criteria (Table 1). To minimize confounding factors (e.g., age and gender), demographics and medical records (e.g., cancer or other disease status and cancer staging and subtypes) were also collected for the metabolomics analysis.

### 2.2. Clinical Sample Collection, Processing, and Storage

Trained nurses collected fasting blood samples from inpatients and outpatients, as described in SOP (Appendix A). Briefly, 5 mL of peripheral blood was collected and kept at 4 °C until pre-processing. Samples were centrifuged at 3800 rpm at 4 °C for at least 10 min, and the supernatant was aliquoted into 2 mL sterile cryopreservation tubes. One aliquot of 1 mL was collected for the biospecimen bank, whereas two aliquots of 100 μL each were collected for metabolite analysis. All aliquots were stored at −80 °C before they were transported to Metanotitia Inc. (Shenzhen, China) for metabolomics analyses. 

### 2.3. Metabolomics Sample Preparation 

Metabolite extraction was performed from an aliquot of 50 μL of each sample using 700 μL of extraction buffer containing methanol/methyl *tert*-butyl ether/water (*v*/*v*/*v*, 1:3:1) and a mixture of chemical standards (gibberellic acid A3: 0.45 μg/mL, ^13^C sorbitol: 1 μg/mL and PC(17:0/14:1): 0.45 μg/mL) as previously described in [16]. Samples were sonicated in a 4 °C bath for 15 min, followed by the addition of 350 μL methanol/water (*v*/*v*, 1:3). Samples were then centrifuged at 12,700 rpm (Centrifuge 5430R, Eppendorf, Germany) at 4 °C for 5 min. The upper phase containing lipophilic metabolites (350 μL) was transferred to a fresh 1.5 mL microcentrifuge tube. The hydrophilic phase was divided into two 2 mL fresh microcentrifuge tubes containing 350 μL and 1000 μL separately, which were further used for GC-MS and C_18_-UPLC-MS based metabolites analysis, respectively. All aliquots were dried using a speed vacuum concentrator without heating and stored at −80 °C for further analysis.

### 2.4. Quality Control (QC) Settings 

To ensure the quality of the measurements and the equipment performance, three types of QC samples were set and prepared following the same procedure as biological samples. The first type of QC encompasses biological samples, named as QC_bio_, in which 50% of randomly selected plasma samples obtained in this study were pooled, or NIST SRM 1950 plasma (QC_NIST_) [17]. The second type of QC samples are a mixture of chemical standards. They are QC_mix_ for GC-MS [18] and LC-MS polar [19] separately and LIPIDOMIX^®^ for LC-MS lipid (Appendix A). The last type of QC contains only solvents, namely, QC_blank_.

Each run of the analytical batch started with a QC_blank_ to detect the impurities in the solvents or the contamination of the separation system. Then, a gradient of QC_mix_ (containing the following concentrations of 40 μg/mL, 27 μg/mL, 20 μg/mL, 13 μg/mL, 10 μg/mL, and 7 μg/mL) was used to fit linear calibration curves and test the MS performance along with concentration. The QC_mix_, QC_bio_, and QC_NIST_ samples were typically analyzed every 10 biological samples. The chemical standards in the QC_mix_ were analyzed with TraceFinder 4.0 software (Thermo Fisher Scientific, Waltham, MA, USA) to monitor the machine and sample status. Before the MS run, the injection order of all biological samples was randomized by hospital (NCH, XYH, TJH), disease group (HCtrl, LC, GC, CRC, and NC), gender (male and female), and age group (under 55 years, 55 to 65 years, and 65 years and older) and then split into batches of 40 samples.

### 2.5. GC-MS and UPLC-MS Detection

The derivatization of the dried polar fraction was done according to Lisec et al. [20]. For each sample, 1 μL was then injected into an Agilent 7890B gas chromatograph with an Rxi^®^-5SilMS GC column (30 m, 0.25 × 30 mm, 0.25 μm) coupled to a Pegasus BT time-of-flight (TOF) mass spectrometer (Leco Corp., St. Joseph, MI, USA) with an electron ionization (EI) source by an Agilent 7683 series autosampler (Agilent Technologies GmbH, Waldbronn, Germany). High-purity helium was used as the carrier gas at a flow rate of 1 mL/min. The temperature was initially set at 50 °C for 2 min and then increased by 1 °C per min until reaching 330 °C. The interface and ion source temperatures were adjusted to 280 °C and 250 °C, respectively. The detector voltage was maintained at 1.2 kV with standard 70 eV EI parameters. 

UPLC-MS analysis was performed using a Waters ACQUITY (Milford, MA, USA) ultra-performance liquid chromatography (UPLC) system coupled to Thermo-Fisher Q-Exactive (Bremen, Germany) mass spectrometers with an electrospray ionization (ESI) source. For the analysis of polar metabolites, the dried polar fraction was resuspended in 200 μL water, and 3 µL of supernatant was transferred to a Waters ACQUITY FTN autosampler with the temperature set to 10 °C. For the lipidomic analysis, the dried lipophilic fraction was resuspended in 200 μL acetonitrile/isopropanol (*v*/*v*, 7:3) solution, and 2 μL of supernatant was transferred to a Waters ACQUITY FTN autosampler with the temperature held at 10 °C. Data were acquired both in positive and negative modes with parameters as described in Appendix A.

### 2.6. Mass Spectrometry Data Processing 

Chromatograms obtained from GC-MS analysis were first exported to NetCDF files using Leco ChromaTOF (version 5.40). Peak detection, fatty acid methyl esters (FAME)-based retention time alignment, and mass spectral comparison with the Fiehn reference libraries [21] were carried out using the Bioconductor package *TargetSearch* [22] in R (version 4.0.3). Metabolite identification was validated through manual inspection of chromatograms. Metabolites were quantified by the peak intensity of a selective mass. Retention time (RT) deviation in GC-MS was calculated as the RT shift throughout all batches for each FAME internal standard (Appendix A).

LC-MS chromatograms were processed using Metanotitia Inc. in-house developed software *PAppLine*^TM^, which was complemented by commercial software including Compound Discoverer 3.1 (Thermo Fisher Scientific, Waltham, MA, USA) and LipidSearch (Thermo Fisher Scientific, Waltham, MA, USA). The parameter settings for the in-house software referred to the commercial software. The first step was to extract peaks from the entire spectrum for each chromatogram and then to apply filtering and a baseline correction algorithm to remove baseline noise while preserving the peaks from the raw signals. The raw continuous data was converted into centroided discrete data. After identifying peaks in individual samples, they were matched across samples and retention time alignment was performed as previously described [23]. The dataset was further refined by removing isotopic peaks, in-source fragments of analytes from ionization, and a lower intense adduct of the same analyte as described by Giavalisco et al. [24,25]. The features were annotated with manual supervision using the Metanotitia Inc. library, Ulib^MS^. Hydrophilic metabolites were annotated via a six-thousand compounds sub-library, which were developed under the same chromatographic and spectrometric conditions as the measured samples, and the lipophilic metabolites were annotated with a sub-library of 1700 lipids and sugar esters that were putatively identified based on the precursor *m*/*z*, fragmentation spectrum, and elution patterns. Elution profiles for each ion group that is likely to originate from the same compound were reviewed. The matching criteria were RT within a window of 0.2 min and *m*/*z* below 10 ppm. Metabolites were quantified via the peak intensity. RT deviation was calculated as: the observed RT minus the expected RT for each chemical standard in QC_mix_ (for LC-MS polar) or in LIPIDOMIX ^®^ (for LC-MS lipid). Duplicated annotations across platforms were removed on the basis of metabolites’ physicochemical properties (i.e., solubility, polarity, RT) and performances (i.e., abundance, intensity, peak shape, annotation score). 

### 2.7. Data Normalization

All annotated metabolites and unknown features with an overall coverage above 50% were selected for further normalization using Python 3.7.6 Anaconda Edition (Anaconda Software Distribution, 2016). The missing values were assigned according to the sample type and the peak abundance. For example, the batch median intensity of the biological samples was used for imputing QC_bio_ and QC_NIST_ missing values. For biological samples, according to features intensity and abundance, different imputation strategies were applied under certain circumstances. For imputation of a feature with an abundance > 50%, a feature median with ±5% random noise was used. For those with an abundance < 50% and a mean intensity > 1E5, platform-specific limit of detection (LOD) with ± 5% random noise was used. For the rest of the features with an abundance < 50% and a mean intensity < 1E5, 0 was used. Batch correction was performed using the QC-based deep learning method implemented in the Normalization Autoencoder (NormAE) package [26]. A non-linear autoencoder was used to remove batch effects. It embeds original high-dimensional data into low-dimensional space that contains biological information without an inter-batch effect. Data reconstruction was performed subsequently to transform those representations in low-dimensional space back into high-dimensional data to remove the inter-batch effect. Furthermore, adversarial regularization and discriminators were used for optimization of the autoencoder to identify and remove the intra-batch effect. QC_NIST_ was used for data normalization in the LC-MS and GC-MS platforms. An optional step was used for scaling the data after merging the data from different platforms [27]. 

### 2.8. Data Analysis with Machine Learning

To classify different groups of samples based on the features of the data matrices, machine learning was performed using different tree-based algorithms including random forest, AdaBoost, and XGBoost [28], with the help of the Python package scikit-learn [29]. Then, 75% of the samples were created as a training data set with stratified sampling, while the remaining 25% of the samples were treated as the testing dataset. The successive halving strategy was used to search for promising candidate combinations of hyperparameters for the XGBoost model [29]. Then, to estimate hyperparameters performance, a 10-fold cross-validation was carried out, dividing the training dataset into ten equal partitions. In each cross-validation, one partition was used as the validation set, while the remaining ones were considered as the training set. The best combination of hyperparameters was determined when the F1 score (F1 Score = 2 × (*Precision* × *Recall*)/(*Precision* + *Recall*)) reached the highest performance. Finally, the F1 score was used to report classification performance in the testing set. 

## 3. Results

### 3.1. Fasting for Blood Sampling

One of the factors that could affect the quality of the plasma is the fasting state. Previous studies have revealed that the reproducibility of several metabolite classes could be lower in non-fasting individuals compared with fasting individuals, which impacts the discovery of new biomarkers for certain diseases [30,31,32]. To elucidate the influence of fasting status on metabolomics comprehensively, we set up a fasting experiment before the cancer cohort study. This investigation was performed with 12 healthy individuals from Metanotitia Inc. to self-compare their blood samples under fasting and non-fasting conditions. We annotated 272, 441, and 86 metabolites from LC-MS polar, lipid, and GC-MS platforms respectively (Figure 2A, Source data 1). These compounds covered different metabolic classes, including amino acids, carbohydrates, nucleotides and their derivatives, bile acids, vitamins and cofactors, xenobiotics, and other organic compounds as well as several lipid classes. To avoid imbalanced weight induced by magnitude differences between platforms, we combined and standardized features across all three platforms by scaling them to unit variance before conducting the principal component analysis (PCA).

As we know, it is often difficult to control the fasting state of the patients in clinical analysis. We performed a principal component analysis to assess the impact of these two feeding states on the global metabolome (Figure 2B). The first principal component (PC1) separated the fasting and non-fasting groups, explaining 13.99% of the variation. Inspection of the loading responsible for the PC1 showed that amino acids and TAGs largely contributed to the discrimination. The PC2 was mainly explained by the variability among the individuals (12.81%). Three well-known markers for fasting [33], namely, 3-hydroxybutyric acid, O-acetyl-L-carnitine, and L-octanoylcarnitines, were also significantly accumulated in the samples of individuals under fasting conditions, whereas all three branched-chain amino acids (BCAAs), i.e., valine, leucine, and isoleucine, were depleted (Figure 2C). Furthermore, the pathway enrichment analysis unraveled the accumulation of intermediates involved in lipid metabolism, such as unsaturated fatty acids, linoleic acid, and alpha-linolenic acid, as well as glycerolipids (Figure 2D). Despite the small number of samples, these results confirmed the importance of having samples on the same fasting status, as diet has a significant impact on the human metabolome, which may interfere with the diagnostics prediction of certain diseases.

### 3.2. A Cancer Pilot Study 

Clinicians at all participating sites followed the same biospecimen collection SOP and kept a detailed sample collection log to guarantee the completeness and accuracy of the data (Figure 1). The median process time, defined as the time from specimen collection to storage at −80 °C, was 53 min. The medical records were used to exclude patients who met any exclusion criteria (Table 1). In addition, the sample collection log was used to rule out the samples that failed to pass the sample quality control. For instance, a sample was discarded due to visible hemolysis, discoloration, too long processing time (>4 h), insufficient volume (<60 μL), and improper storage.

Among 2490 consenting subjects, 190 were excluded because of either meeting one of the exclusion criteria or failing in the sample quality check. More specifically, 4 subjects were excluded as they are below 18 years old; 12 samples were repeatedly enrolled. A total of 156 participants exhibited unconfirmed diagnoses, and 18 samples were wiped out because of failed quality checks. Out of 2300 eligible participants, 1292, 772, and 236 subjects were enrolled in the cancer, non-cancer, and healthy group, respectively (Figure 3). The cancer group was further divided into LC, GC, and CRC sub-groups and a sub-group of other cancers. 

The demographic information is shown in Table 2. Differences were observed based on sample size, age, female percentage, chronic diseases, smoking and alcohol-drinking history between healthy control, cancer, and non-cancer groups. These factors may contribute to interindividual variations in the measurements. 

### 3.3. Metabolomics Analysis and Quality Control

We used a simple, robust, and repeatable two-phase liquid-liquid extraction process, which can extract polar, semi-polar, and hydrophobic metabolites as well as proteins from a single sample, therefore, it is suitable for high-throughput analysis [24]. The extraction was manually performed in 70 batches for all samples. A maximum of three batches can be analyzed weekly with a single GC-MS and two LC-MS platforms running in parallel. In total, 3762 samples were analyzed in 28 weeks, including 2491 biological plasma samples from 2300 subjects, 123 QC_blank_, 277 QC_NIST_, 279 QC_mix_, and 592 QC_bio_ samples.

The average intra-batch CV of the spiked-in chemical standards is less than 13%, which indicates a stable performance in metabolite extraction. To assess the performance of our analytical workflow, we included several QCs during the process to track possible contaminants and MS performance. As summarized in Figure 1, such QCs can help to correct systematic error. The mass error (Mass_err_), RT deviation (RT_dev_), and the intensity coefficient of variations (Int_CV_) were monitored via chemical standards in QC_mix_ and FAME, for LC-MS and GC-MS, respectively. 

Ideally, the entire analytical window should include fully distributed analytes across the *m*/*z* and RT range. Here, we present comprehensive QC reports for plasma samples as an example of good practice. Detailed QC results include the mean RT_dev_ and mean Mass_err_ and Int_CV_ of each QC_mix_ standard from both LC-MS platforms, as well as the mean RT_dev_ and Int_CV_ of each FAME standard and QC_mix_ from the GC-MS platform. These QC parameters provide a global view on the MS performance, which facilitates us to assess the analytical methods and quality of MS data. As shown in Figure 4 and Appendix A, the mean RT_dev_ ranged from −0.1 to 0.15, −0.1 to 0.26 and −0.1 to 0.1 for LC-MS polar, LC-MS lipid, and GC-MS, respectively, and the extra-batch Int_CVs_ were below 30%. The mean Orbitrap Mass_err_ of the LC-MS platform was below 10 ppm (polar) and below 5 ppm (lipid; Figure 4A,B, Appendix A). Altogether these results confirmed that all our three analytical MS platforms were running stably in large batches spanning a long period, producing highly reliable data for further analysis. 

### 3.4. Data Normalization

With the processed chromatograms of 3762 samples, 4380, 5415, and 92 features were yielded from LC-MS polar, LC-MS lipid, and GC-MS platforms, respectively. Among those, 504, 611, and 92 features were annotated (Source data 2). Unknown features with an overall coverage < 50% were excluded for further analysis, resulting in 2581 and 2575 features for polar and lipid analysis, respectively. The COVID-19 outbreak in early 2020 delayed the analysis of the plasma samples for three months. Consequently, data acquisition for the cancer pilot study was separated into two phases, imposing challenges in data normalization. To gain a good normalization performance, we compared several methods, such as total ion count (TIC) [34], the median of feature intensity, and a deep learning method, NormAE, that calibrates nonlinear batch effects [26,35].

As QC_NIST_ was identical throughout the entire dataset and the variation only came from the day of injection, this sample could be an excellent proxy to evaluate and normalize the batch effects. Therein, we presented intensities of representative features before and after normalization (Appendix A). In comparison to the TIC and median normalization methods, NormAE significantly reduced the unexpected systematic variations among samples. This is evidenced by the flattened intensity variation without a visible inter-batch or intra-batch effect. We further tested the normalization performance of NormAE with PCA analysis in QC_NIST_ and biological samples (Figure 5). Before normalization, even though the QC_NIST_ samples tended to group together, they were still spread with the biological samples in all three platforms, indicating the presence of batch effect (Figure 5). After NormAE normalization, QC_NIST_ samples were tightly clustered and further separated from the biological samples. NormAE allows a satisfying normalization performance in the case of data acquisition over a long period.

### 3.5. A Case Study of Colorectal Cancer Sub-Cohort

Colorectal cancer (CRC) is the second most common adult cancer in women and the third most common in men, and it is the fourth leading cause of cancer death [36,37]. Age and genetic and environmental factors play a major role in the development of colorectal cancer [38]. Hence, studying and recognizing CRC disease progression and patterns via metabolomics will provide direct evidence of the outcome from the interaction between genes and environment. Evaluating the metabolic profile of CRC is gaining popularity in clinical metabolomics studies. CRC possesses significant perturbations in cellular respiration and metabolism, i.e., carbohydrate, lipid, amino acid, nucleotide, and ketone metabolisms [39], which could provide a reference for our data mining and analysis. Here, we selected CRC metabolic data for a case study, which forms a cohort from our cancer pilot study. 

The CRC sub-cohort was composed of 236 healthy control (HCtrl) and 279 CRC samples (Appendix A). Samples of both groups were collected from independent centers and distributed evenly in 70 batches during data acquisition, indicating a good representation of the entire cancer cohort study. In total, we identified 5211 features from three MS-based platforms. After rigorous filtering and deduplication, there were 1109 features annotated. In comparison to the HCtrl group, 55 and 62 metabolites were significantly increased and decreased in CRC samples (Figure 6A). Therein, 82 metabolites have been reported as potential biomarkers of CRC and 10 of them were significantly changed in our study, including hypoxanthine, palmitoleic acid, cis-aconitic acid, fatty acid 20:2, elaidic acid, gondoic acid, pyruvic acid, O-hydroxyhippuric acid, trans-4-hydroxy-L-proline, and tryptophan (Figure 6A). Overrepresentation analysis of these metabolites among metabolic pathways unravels a major enrichment in the metabolism of lipids and amino acids (Figure 6B). 

Appropriate classification methods are critical in predictive analysis to uncover links between potential biomarkers, complex phenotypes, and clinical data. In this study, three tree-based algorithms, namely, AdaBoost, random forest (RF), and XGBoost, were selected for classification. Their performance was evaluated via an F1 score based on a 10-times repeated trial for each [40]. Compared to the other two methods, XGBoost produced the tightest cluster with the highest (>0.95) F1 scores for both HCtrl and CRC classes, which suggests that XGBoost is potentially useful for the classification of different groups of metabolomics samples (Figure 7).

## 4. Discussion

Metabolomics has been practiced for several decades and unveiled crucial biological information in fundamental research [41]. Nonetheless, there are still challenges in applied aspects, especially in clinical practice. Metabolomics is one of the most promising technologies for precision medicine. It is particularly important to develop a comprehensive and easy-handling pipeline that includes all critical steps with detailed descriptions from clinical sampling, (pre-)analytical handling, metabolomics data acquisition, and processing. This study fully utilized the advantages of metabolomics while circumventing its limitations in the context of the actual clinical situation. It provides clear guidelines for clinicians and researchers to achieve accurate results for diagnosis and therapy. 

### 4.1. Factors That May Influence Plasma Metabolomics Analysis

Because of the physicochemical diversity of metabolites, particularly in the dynamic range of abundance, it is critical to have the samples in a proper status. This could reduce noise as much as possible while ensuring sample quality. In comparison to genomics studies, the metabolic samples are more sensitive to individualized activities such as diet and pre-analytical handling (i.e., sensitive to temperature/light and continued in vitro enzymatic reaction) [15]. Hence, we emphasized specific sampling suited for clinical metabolomics studies. For example, it is recommended that blood samples be collected from fasted subjects to minimize individual diversity because of the strong perturbation of ketone body, BCAAs, and acylcarnitine metabolism due to fasting (Figure 2). Moreover, the quality of plasma samples subjected to metabolomics analysis is determined by the handling time and temperature before the freezing of samples in liquid nitrogen or at −80 °C [42]. Blood samples should be transported using 4 °C pre-cooled boxes and processed at 4 °C in the clinic lab (Appendix A). This pre-analytical handling should be finished within two hours. Time and temperature not only matter before the pre-analytical phase but also during storage. In one metabolomics study, the human blood sample was stored for 16 years, and the majority of the metabolites remained stable within the first four years at −80 °C [43]. However, they only investigated targeted metabolites with GC-MS, which may not be able to represent the whole metabolome. Another ^1^H-NMR chemometrics study in human blood only tested metabolic stability up to 31 days at −80 °C [44]. In this study, although pre-processed samples were frozen at −80 °C until the day of data acquisition, it is still unknown whether prolonged storage can affect the metabolome. A systematic investigation of the effect of long-term storage on the quality of metabolic samples could be considered. Here, we provided a practical example of balancing the trade-off between long storage time and consecutive data acquisition in a large-scale study. Despite the data acquisition being interrupted by the COVID-19 lockdown, the proposed normalization results were satisfying (Appendix A).

### 4.2. Good Practices for Large-Scale Data Acquisition and Processing

In this work, we proposed several data acquisition and data processing strategies to guarantee the quality of metabolomics data for downstream analysis.

First, proper sample randomization and enough QC sample replicates were included to ensure optimal statistical power in the downstream bioanalyses. In this study, 2491 blood samples were randomized as described in Section 2.4 and then analyzed with 1271 QC samples (123 QC_blank_, 277 QC_NIST_, 279 QC_mix_, and 592 QC_bio_), each of which served critical functions. QC_blank_ samples were used to detect contamination across the workflow. QC_NIST_, the popular pooled blood samples recommended in other metabolomics studies [18,45,46], were employed for removing batch effect. The FAME mix and QC_mix_, which consists of chemical standards evenly distributed across the RT, were used to monitor chromatographic and mass spectrometric stability as well as the subsequent RT correction. The QC_bio_ samples pooled by healthy and cancer subjects were used to improve annotation by avoiding potential ion suppression. They also served as a cost-effective alternative to analyzing all samples in MS/MS. 

Second, as machine maintenance routines and inevitable machine fluctuation can both affect the machine sensitivity, a machine log was generated for each acquisition date so that operators could identify the source of a problem when it appeared (e.g., an abnormal mean intensity gap between batches). 

Third, we chose QC_NIST_ over QC_bio_ as a sustainable normalization strategy for large-scale studies. In our framework, there were three major challenges for data normalization: (1) Despite the fact that using multiple spiked-in internal standards for subsequent normalization has many advantages [47,48], the metabolite ionization and response might be influenced by the matrix ion suppression or enchantment. This strategy is not applicable for unknown features [49]. (2) We implemented several scaling factor normalization methods, including median and total ion count (TIC). As metabolites are physiological end-points that can be largely affected by the environment and lack a self-averaging property [47], these normalization methods did not show satisfactory results (Appendix A). (3) Due to the enormous number of samples needed in the clinical studies, waiting until every sample is collected to make a pooled sample will significantly delay operation progress, add to the financial burden, and require more storage resources with an increasing number of samples. Thus, using QC_NIST_ for normalization enables parallel sample collection and acquisition regardless of the timespan. In this study, we adapted a recently developed algorithm, the normalization autoencoder (NormAE), to remove inter- and intra-batch effects [26]. As artificial neural networks are commonly used to uncover complex patterns in the data [50], such techniques can distinguish the latent representations that contain biological information from batch effects [26]. Although concerns have been raised in recent years about the danger of a “black box” in the output interpretation in AI-related works [51], we are confident in our data normalization as it was validated via the QC_NIST_ samples serving as the proxy of biological samples and demonstrated high homogeneity after normalization (Figure 5). 

In addition, during data processing, we combined the features across all three platforms and did deduplication of the metabolites detected via multiple analytical platforms, which was conducted based on existing knowledge: (1) GC-MS data were preferred for sugars, amino acids, hydroxy short-chain fatty acids, TCA cycle-involved metabolites, urea, uric acid, creatinine, and creatine when these features were detected by both LC-MS and GC-MS. (2) For long-chain fatty acids with more than 10 carbons, LC-MS lipid data were preferable because they are more soluble in the lipophilic phase. (3) Metabolites with an RT between 2 and 14 min were preferred for duplication across LC-MS polar and lipid platforms. Despite the above criteria, annotations with better peak shape, higher mean intensity, and higher abundance are favored. For example, the GC-MS data were preferred for α-ketoglutaric acid when it was detected by both LC-MS polar and GC-MS platforms because (1) it is a TCA cycle-involved metabolite; (2) its RT in the LC-MS polar platform was before 2 min; (3) although its abundance in GC-MS is lower than in LC-MS polar (70% vs. 80%), its annotation score is higher. Another example was hydrocortisone; LC-MS polar data were preferred when it was detected by both LC-MS polar and LC-MS lipid platforms, with higher mean intensity, preferable RT (2.8 vs. 0.7 min), and higher abundance (95% vs. 46%). Furthermore, some metabolites exhibited different change trends on different platforms. For example, 3 out of 1109 metabolites of the CRC case study possessed different perturbation trend on polar and lipid platforms. In addition to the above criteria and methods, we selected the platform data by assessing the significance of change on different platforms (i.e., foldchange and *p* values). Thereafter, the cleaned data was used for statistical and biological analysis. 

### 4.3. Biomarker Evaluation

Biomarkers have revolutionized research on diverse diseases, particularly cancers, drug trial design, and patient management in clinical practice. One of the most challenging issues in untargeted metabolomic biomarker discovery is reproducibility, which may be caused by cohort-related factors, pre-analytical and analytical factors, biotemporal variability, improper statistical analysis, and insufficient validation [52]. In the case study of fasting, we obtained a good reproducibility of fasting biomarkers, 3-hydroxybutyric acid and two acylcarnitines (Figure 2C). A previous study set fasting periods of 10, 34, and 58 h for four volunteers and conducted a longitudinal comparison [53]; however, our blood samples were collected after only 9 h fasting and compared with non-fasting samples. Converse to the previous study [53], the depletion of BCAAs in our fasting samples may be due to different fasting status resulting in different degrees of metabolic reprogramming, as well as a different comparison baseline. Interestingly, we noticed that the metabolism of unsaturated fatty acids and glycerolipids was upregulated under fasting (Figure 2D). Linoleic acid, as the most highly consumed fatty acid in the human diet, functions as a membrane component and can also be used as a source of energy [54]. Alpha-linolenic acid was assumed to be a precursor of long chain fatty acids, eicosapentaenoic acid (EPA) and docosahexaenoic acid (DHA), whereas its major metabolic route is beta-oxidation [55], which could provide energy for bioactivities. This information indicates the potential relationship between lipid metabolism and fasting. 

Although advances in colorectal cancer (CRC) treatment have led to a radical rise in survival over recent decades, identifying predictive and/or prognostic biomarkers remains a challenging issue. In addition to gene mutations and alterations in ncRNA, dysregulations of metabolism could also be indications of CRC progression. As reported previously, CRC occurrence is usually associated with a wide effect on metabolic pathways, including urea cycle, ammonia recycling, protein biosynthesis, alanine metabolism, citric acid cycle, and glutathione metabolism, and hundreds of metabolites involved in CRC progress have been identified [39,56,57,58]. In our CRC sub-cohort study, we identified 82 metabolites targeting the summarized CRC biomarker list, 50 of which showed change trends that are consistent with those reported in the literature (i.e., 32 and 18 metabolites were accumulated and depleted respectively), and 10 of them were significantly changed (Figure 6A). Due to differences in analytical platforms, study design, and disease stages, most dysregulated metabolites of CRC identified in different studies were inconsistent. Comparatively, the screening of our workflow was more robust, with high coverage and reproducibility. In this methodological paper, the CRC sub-cohort just served as an example that facilitates us to roughly assess the performance of the entire workflow and the reliability of metabolomics results. Thereafter, the biomarker panels of CRC will be reproduced and validated precisely and developed as a licensed diagnostic product. 

### 4.4. Limitations and Outlook

QC_NIST_ is a pooled sample collected from representative races with both genders, including 100 fasted healthy individuals in the United States. We discovered that there are many QC_NIST_-specific as well as our healthy control (HCtrl)-specific features, which may be due to the difference in the genomic and environmental background of subjects. It would be more scientifically comparable as well as more sustainable to generate a QC_NIST_ sample from the region of interest in the cohort location under the guideline of QC_NIST_ generation and certification. Moreover, the injection order of all biological samples was randomized by hospitals, disease groups, age, and gender and evenly divided into 70 batches to avoid batch and center effects confounding the biological differences. This approach is sufficient for measurements with ready cohort samples. Moreover, a metabolomics-based approach for broadly applicable cancer screening necessitates very large-scale, potentially population-level clinical evidence for both test development and clinical validation. 

To adequately characterize the cancer-defining biomarkers, at least several hundred cases are required for each cancer of interest. Given that most cancers were detected at a late stage, a larger cohort is expected to include cancers with minor subtypes or at early stages. Metanotitia Inc. is conducting several studies to profile the metabolic signatures of multiple diseases, including cancer and chronic diseases, and the background patterns of matching healthy subjects. In this regard, Metanotitia Inc. is developing a next-generation pipeline designed for parallel sample collection and metabolomics profiling, laying the technical groundwork for the development of a multi-disease diagnostic test at a population level. 

## 5. Conclusions

We developed a workflow for clinical metabolomics, which has the following advantages. (1) The well-practiced SOP for sample collection and transportation from clinical sites assured the sample quality. (2) The combination of three MS-based platforms facilitated the detection of metabolites, broadening the chemical space coverage and increasing the likelihood of discovering biomarkers. (3) Up to 50% of QC sample settings enabled us to monitor the MS performance systematically and assisted the following data processing. (4) The compound identification was made through annotation based on both commercial and in-house-developed software and a library. Moreover, the entire routine was described and validated by case studies to demonstrate its reliability and reproducibility. In a word, our clinical metabolomics work frame demonstrates reproducible good practice and workflow, which can tutorize especially the applications of big data-driven metabolomics in clinical studies.

## Figures and Tables

**Figure 1 metabolites-12-01168-f001:**
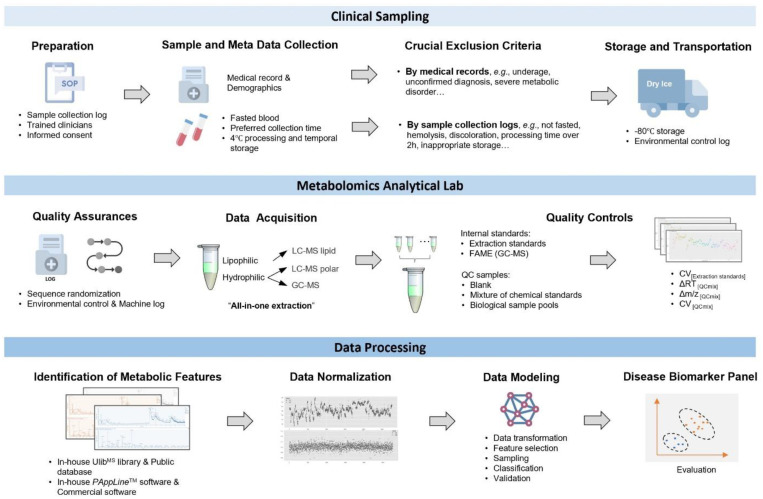
The workflow for a clinical-oriented metabolomics study.

**Figure 2 metabolites-12-01168-f002:**
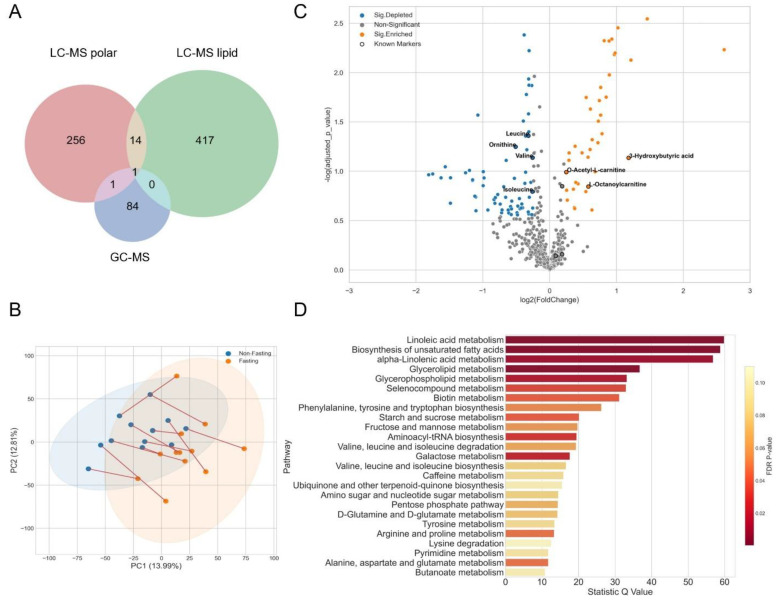
Metabolic profiles and comparisons of plasma samples from identical subjects in the fasting and non-fasting status. (**A**) Venn diagram of annotated metabolites from GC-MS and LC-MS (polar and lipid) platforms. (**B**) The PCA analysis of metabolic profiles of fasting (orange dots) and non-fasting (blue dots) samples from identical subjects (connected by red lines). The eclipses are the 95% confidence intervals. (**C**) Volcano plot of significantly changed metabolites on fasting status. |Log2(foldchange)| > 0.25, *p* value < 0.05. Orange dots: significantly accumulated metabolites; blue dots: significantly depleted metabolites; gray dots: non-significant; black circles: the well-known fasting biomarkers from literature. (**D**) Pathway enrichment analysis from significantly changed metabolites by comparing the fasting and non-fasting status. Visualization was based on MetaboAnalyst 5.0 (https://www.metaboanalyst.ca/ (accessed on 18 August 2022)).

**Figure 3 metabolites-12-01168-f003:**
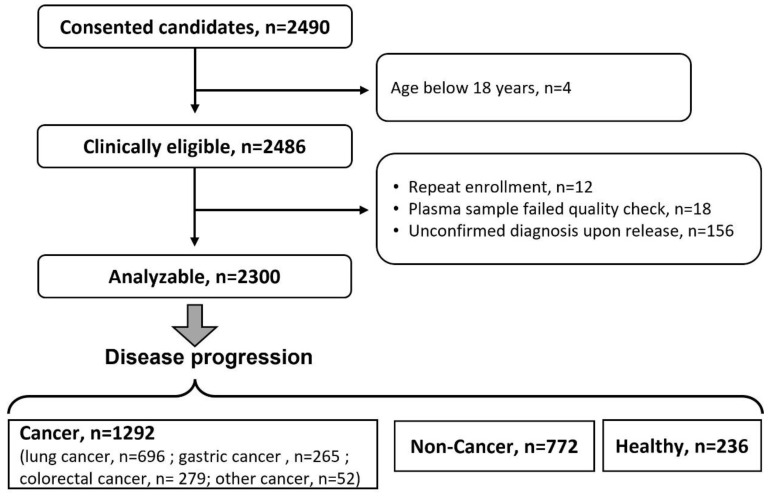
Flow chart of the patient disposition in the cancer pilot study. A total of 190 were excluded by either meeting one of the exclusion criteria or failing the sample quality check, and 1292, 772, and 236 patients were enrolled in the cancer, non-cancer and healthy group, respectively.

**Figure 4 metabolites-12-01168-f004:**
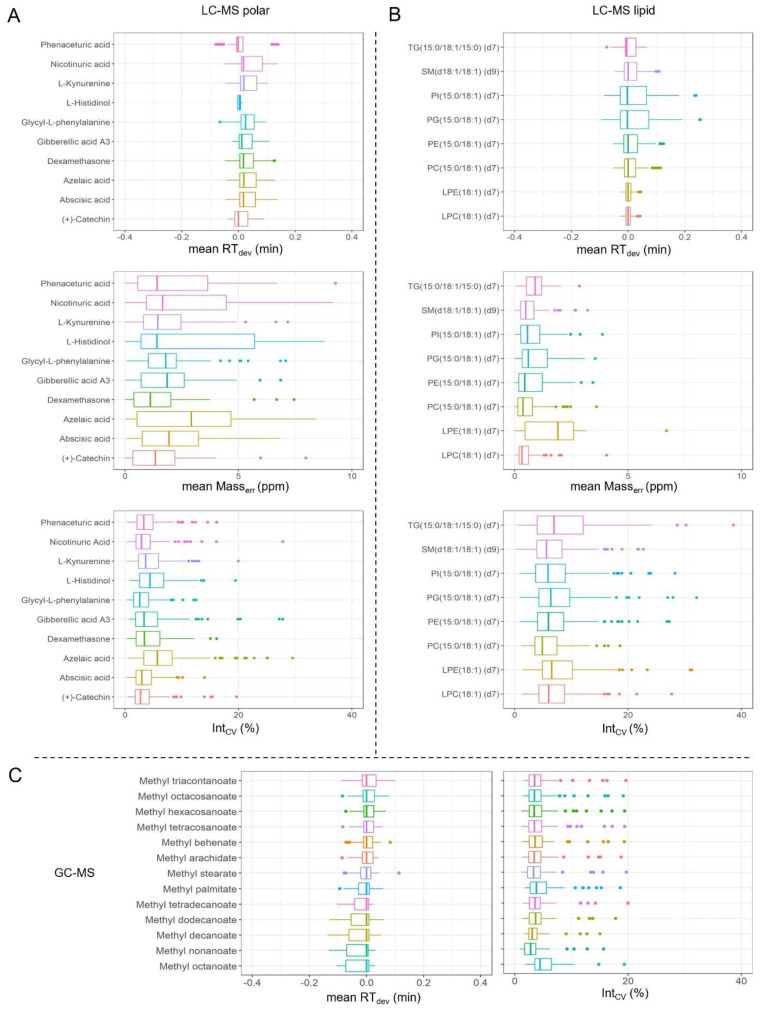
The performance of MS-based platforms across the whole study was visualized using box plots of variable parameters observed from reference compounds. (**A**) LC-MS polar platform. (**B**) LC-MS lipid platform. (**C**) GC-MS platform. RTdev: delta retention time (RT), Masserr: mass error, IntCV: intensity coefficient variation. Each standard value was calculated from 70 batches.

**Figure 5 metabolites-12-01168-f005:**
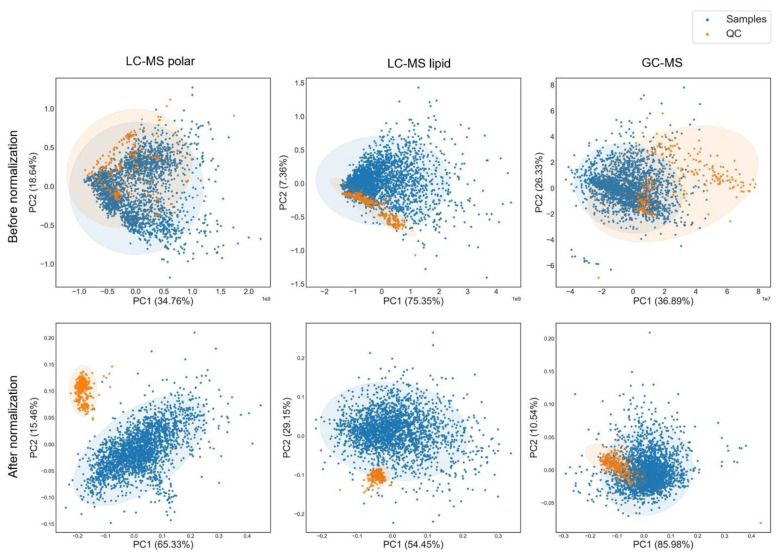
The normalization performance of NormAE. The PCA plots show the normalization performance of the NormAE method [24]. Orange dots: QCNIST; blue dots: biological samples.

**Figure 6 metabolites-12-01168-f006:**
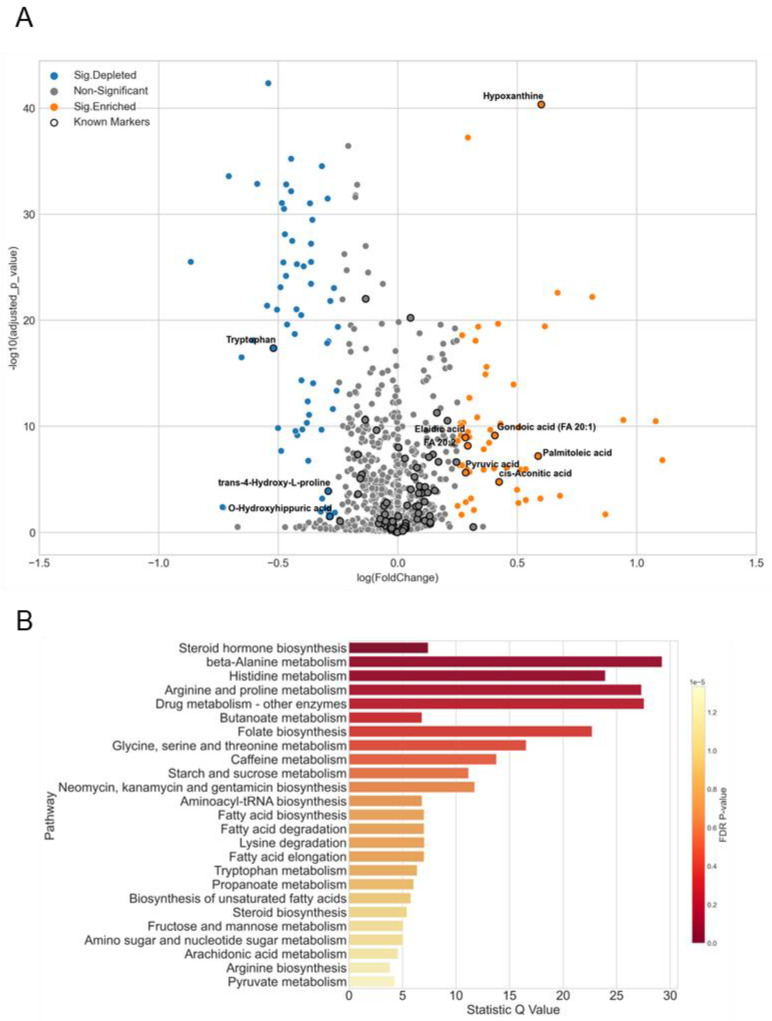
Differentiated metabolites in CRC samples. (**A**) Volcano plot of significantly changed metabolites in CRC. |Log2(foldchange)| > 0.25, *p* value < 0.05. Orange dots: significantly accumulated metabolites; blue dots: significantly depleted metabolites; gray dots: non-significant; black circles: reported biomarkers of CRC from literature. (**B**) Pathway enrichment analysis from significantly changed metabolites by comparing CRC and healthy control groups. Visualization was based on MetaboAnalyst 5.0 (https://www.metaboanalyst.ca/ (accessed on 11 September2022)).

**Figure 7 metabolites-12-01168-f007:**
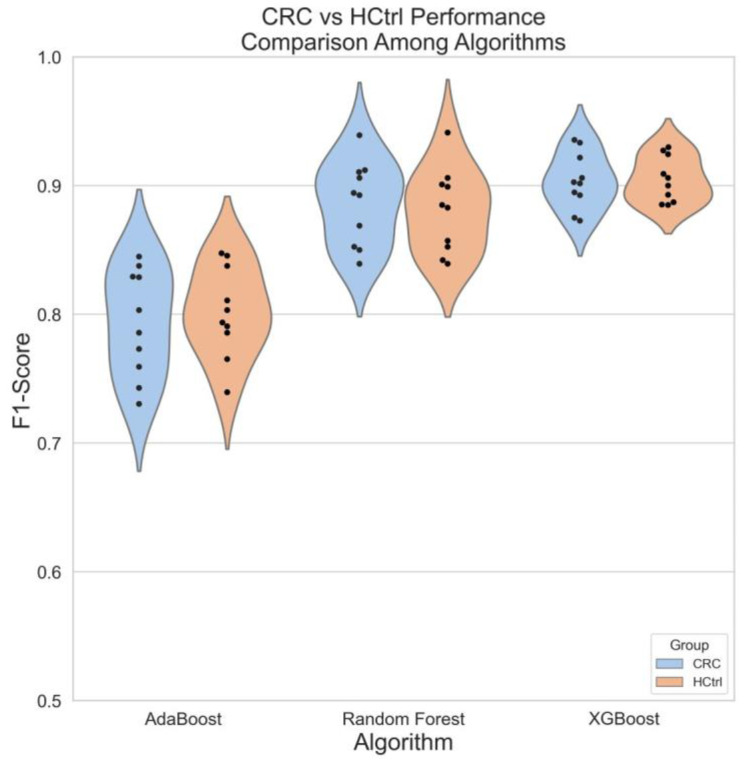
Violin plots from the F1 scores of the classification models from CRC patients and healthy control subjects. Three tree-based classifier algorithms, AdaBoost, random forest, and XGBoost, were selected for classification. Each model was run 10 times. Orange: healthy control (HCtrl); blue: colorectal cancer (CRC). F1 Score = 2 × (*Precision* × *Recall*)/(*Precision* + *Recall*).

**Table 1 metabolites-12-01168-t001:** Inclusion and exclusion criteria.

Group	Inclusion	Exclusion
Cancer	hospitalized for surgery or screenings at participating clinical sites	younger than 18 years old severe metabolic diseases pregnancy unconfirmed diagnosis require any emergency treatments
Non-cancer	hospitalized for surgery or screenings indication at participating clinical sites diagnosis other than cancer	
Healthy	underwent physical exams at participating clinical sites screened with chest low dose computed tomography (LDCT),	younger than 18 years old severe metabolic diseases pregnancy diabetes hypertension cardiovascular diseases any other abnormalities requiring immediate intervention

**Table 2 metabolites-12-01168-t002:** The demographics of the cohort study.

Demographics		HCtrl	Cancers	Non-Cancer
n		236	1292	772
Hospital, n (%)	TJH	-	57 (4.4)	50 (6.5)
	NCH	128 (54.2)	852 (65.9)	591 (76.6)
	XYH	108 (45.8)	383 (29.6)	131 (17.0)
Mean age (SD)		45.6 (13.4)	60.9 (10.9)	54.6 (13.6)
Female (%)		124 (52.5)	469 (36.3)	309 (40.0)
Diabetes (%)		-	80 (6.2)	37 (4.9)
Hypertension (%)		-	319 (24.8)	123 (16.1)
Smoking History (%)	Never	171 (72.5)	789 (61.9)	533 (70.0)
	Current	57 (24.2)	317 (24.9)	166 (21.8)
	Used to	8 (3.4)	169 (13.3)	62 (8.1)
Alcohol-drinking History (%)	Never	153 (64.8)	1027 (81.1)	599 (78.9)
	Current	83 (35.2)	213 (16.8)	147 (19.4)
	Used to	-	27 (2.1)	13 (1.7)

TJH: The Second Hospital of Tianjin Medical University; NCH: First Affiliated Hospital of Nanchang University (Nanchang, Jiangxi Province); XYH: Second Xiangya Hospital of Central South University (Changsha, Hunan Province); HCtrl: healthy control.

## Data Availability

The dataset used and/or analyzed during the current study are available from the corresponding author on reasonable request as it is crucial to their commercial IP.

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
