# Peer review of "A Comprehensive Mass Spectrometry-Based Workflow for Clinical Metabolomics Cohort Studies"

_metabolites, 2022, doi:10.3390/metabo12121168_

Round 1

Reviewer 1 Report

The work describes  complex metabolomics workflow based on three analytical platforms involving SOPs for sample preparation, instrumental analysis and data processing and mining.  Interestingly, the earlier experience from plant metabolomics was transfered to clinical field.

Is there some limitation in the metabolite coverage, when the proposed workflow is used ? Although the  manuscript is carefully prepared and well organized, the report is sometimes too general and not very informative. 

Major issue:

To prove the relevant performance of the workflow (the robustness), the annotated metabolites (504/611/92 features) with the observed characteristics (RT, m/z (pos/neg ESI HRMS), RSD of the peak area before and after normalization) have to be enclosed for the pilot studies (fasting, CRC). The perturbed metabolites should be commented, particularly those detected by more analytical platforms. How the normalization method was used, it also should be described in a greater detail. 

Author Response

We chose the MTBE method for metabolite extraction, which can extract polar and nonpolar metabolites in a single step. This method has been well practiced and can cover a wide range of metabolites except for extremely polar metabolites, such as sugars and betaine groups (Cajka et al., Anal. Chem. 2016). For data acquisition, we used GC-MS and LC-MS platforms. All these strategies enhanced the metabolite coverage. 

We apply related technologies to animal and disease research (Barbash, Set al., 2017. Neurobiology of disease, 106, 1-13; Li, Q., et al., 2017. Molecular biology and evolution, 34(5), 1155-1166; Bozek, K., et al., 2015. Neuron, 85(4), 695-702.). This study developed a comprehensive workflow for clinical metabolomics cohort studies, especially for disease diagnosis on a population level, annual health examination and management, and clinical trials for disease biomarker discovery and validation. Although this workflow lacks some creativity, every detail described in this manuscript is critical to a successful clinical metabolomics experiment.  Thus, this workflow can tutorize especially the applications of big data-driven metabolomics in clinical studies. 

Due to competing commercial interests, the characterization of the annotated metabolites is not available. Some of these compounds are in product development.  

Deduplication of the metabolites detected by multiple analytical platforms was conducted on the basis of existing knowledge. GC-MS data were preferred for sugars, amino acids, hydroxy short-chain fatty acids, TCA cycle-involved metabolites, urea, uric acid, creatinine, and creatine were detected by both LC-MS and GC-MS. For long-chain fatty acids with more than 10 carbons, LC-MS lipid data were preferable because they are more soluble in the lipophilic phase. Metabolites with RT between 2 and 14 minutes were preferred for duplication across LC-MS polar and lipid platforms. Despite of above criteria, annotations with better peak shape, higher mean intensity, higher abundance and higher annotation scores are favored.

The following cases comprehensively applied the above criteria. The GC-MS data were preferred for alpha-Ketoglutaric acid when it was detected by both LC-MS polar and GC-MS platforms because 1) it is a TCA cycle-involved metabolite; 2) its RT in LC-MS polar platform was before 2 minutes; 3) although its abundance in GC-MS is lower in GC-MS than in LC-MS polar (70% vs. 80%), its annotation score is higher in GC-MS. Another example was hydrocortisone, LC-MS polar data were preferred when it was detected by both LC-MS polar and LC-MS lipid platforms, with higher mean intensity, preferable RT (2.8 vs. 0.7 minutes), higher abundance (95% vs. 46%).

A non-linear autoencoder was used to remove batch effects. It embeds original high-dimensional data into low-dimensional space that contains biological information without inter-batch effect. Data reconstruction was performed subsequently to transform those representations in low-dimensional space back into high-dimensional data to remove inter-batch effect. (line 242 to 246)

Reviewer 2 Report

Section 4.3 Biomarker evaluation.  Authors go into great detail in the introduction to discuss biomarker analysis and the use of metabolomics studies to find biomarkers in clinical studies, but the section is lacking elucidation of their findings.  Certain metabolites are references but no effort is made in explaining their biological role.

Section 4.3 Biomarker evaluation.  No mention of the additional cancer cohorts are mentioned again regarding significant metabolites found and their biological role.

2.6. Mass spectrometry data processing. Use of in-house software used for data processing, but no reference for the software used.  If no reference available, explanation of the software should be included in supplemental info.

2.6. Mass spectrometry data processing.  Data processing parameters are insufficiently explained.

4.1. Factors that may affect the quality of plasma suited for metabolomics analysis. Authors present suggestions for sample handling in clinical setting; however, there is no data in the paper that they show to support these references to temperature control.  Authors need to include their own trials and experimental data or reference literature.

Author Response

Section 4.3 Biomarker evaluation.  Authors go into great detail in the introduction to discuss biomarker analysis and the use of metabolomics studies to find biomarkers in clinical studies, but the section is lacking elucidation of their findings.  Certain metabolites are references but no effort is made in explaining their biological role.

For disease biomarker screening, the most important thing is producing highly reproducible results. The biomarker panel can consist of any biomolecules (gene, protein, metabolites, etc.), clinical characteristics, and demographic factors (e.g., age, sex, BMI). Thus, we have not yet focused on the biological role of what we screened out, but they can help us understand disease mechanisms and judge the reliability of putative biomarkers with reference to previous studies.  

Section 4.3 Biomarker evaluation.  No mention of the additional cancer cohorts are mentioned again regarding significant metabolites found and their biological role.

In this study, we developed the whole pipeline for clinical metabolomics. To assess the performance and reliability, we chose the fasting and CRC cases to do further data analysis. These two cases have been studied in extensive research. Thus, we can compare our results with the previous findings, which facilitates us to improve our workflow if it is necessary and prove the robustness of our pipeline.

2.6. Mass spectrometry data processing. Use of in-house software used for data processing, but no reference for the software used.  If no reference available, explanation of the software should be included in supplemental info.

LC-MS chromatograms were processed using Metanotitia Inc. in-house developed software PAppLineTM to convert the raw data into a peak list. The first step was to extract peaks from the entire spectrum for each chromatogram, then to apply filtering and baseline correction algorithm to remove baseline noise while preserving the peaks from the raw signals. The raw continuous data was converted into centroided discrete data. After identifying peaks in individual samples, they were matched across samples and retention time alignment was performed as described in Smith et al. 2006. Elution profiles for each ion group that are likely to originate from the same compound were reviewed. (line 206-212)

2.6. Mass spectrometry data processing.  Data processing parameters are insufficiently explained.

PAppLineTM was complemented by commercial software including Compound Discoverer 3.1 (Thermo Fisher Scientific, USA) and LipidSearch (Thermo Fisher Scientific, USA). Thus, the parameters for data processing were based on the default of commercial software. (line 206-207)

4.1. Factors that may affect the quality of plasma suited for metabolomics analysis. Authors present suggestions for sample handling in clinical setting; however, there is no data in the paper that they show to support these references to temperature control.  Authors need to include their own trials and experimental data or reference literature.

We added references 15 and 40 to support our suggestions for clinical sampling. As described in these method papers, metabolic sample preparation is particularly vital in a metabolomics workflow. For tissue or cell samples, the rapid stopping, or quenching, of metabolism and extraction of the metabolites is required to get quantitatively stable metabolomics results. So quick freeze, liquid N2 grinding, and clamping are suggested. For biofluids such as serum, plasma, or urine samples, which are relatively low metabolically active systems, cold organic solvents are suggested.

Reviewer 3 Report

Comments to the authors

The authors provide a very well-designed study on mass spectrometry-based workflow for clinical metabolomics cohort studies. However, a few minor issues should be clarified.

Introduction: the authors suggest the use of well-practiced standard operation procedures for sample collection, storage, and preparation but they do not report them in detail. They should be explained the impact of uncorrected sample conditions or manipulations on the quality of the analysis in the state of the art. Please clarify

Line 128: the unit for the centrifuge should be expressed in g, not in rpm. Please correct.

Line 134: the authors describe the use of three different quality controls and the only use of QC mix for calibrations. In my experience, calibrators are usually prepared in the matrix of interest, have the authors considered this aspect?

Line 295: the number of patients is slightly unbalanced (236 control group vs 1292 cancer group). Have the authors considered this limitation in their statistic evaluations?

Author Response

Introduction: the authors suggest the use of well-practiced standard operation procedures for sample collection, storage, and preparation but they do not report them in detail. They should be explained the impact of uncorrected sample conditions or manipulations on the quality of the analysis in the state of the art. Please clarify

For the details of SOP, please check out File S1. To ensure the sample quality, every step should be carefully considered. Because the full study of the possible combinations of all types of preanalytical factors would require huge numbers and volumes of samples, one single research group cannot perform all the necessary testing. Thus, the SOP was conducted for clinical sampling according to our previous experience (Barbash, Set al., 2017. Neurobiology of disease, 106, 1-13; Li, Q., et al., 2017. Molecular biology and evolution, 34(5), 1155-1166. Bozek, K., et al., 2015. Neuron, 85(4), 695-702.), protocols from other literature, and actual situation practices.

The most important factors for keeping the biospecimen quality are storage temperature and handling time (reference: Human Biospecimen Research: Experimental Protocol and Quality Control Tools.  https://doi.org/10.1158/1055-9965.EPI-08-1231). There is a high risk if samples are handled and stored under inconsistent or incorrect conditions. For example, the handling time should be controlled as short as possible to reduce the hemolysis or changes in the component levels. In actual practice, blood samples were collected at the ward of the hospital and transported to the laboratory for processing. Many of our efforts for clinical practice are aimed at controlling the processing time within 2 hours at different clinical sites. In this study, there are crucial exclusion criteria to ensure the sample quality and uniformity by sample collection logs (Fig. 1), which can reduce the failure of clinical metabolomics experiments as much as possible. 

Line 128: the unit for the centrifuge should be expressed in g, not in rpm. Please correct.

Done.

Line 134: the authors describe the use of three different quality controls and the only use of QC mix for calibrations. In my experience, calibrators are usually prepared in the matrix of interest, have the authors considered this aspect?

In this study, QCmix consists of chemical standards evenly distributed across the RT, which is designed for monitoring chromatographic and mass spectrometric stability as well as the subsequent RT correction. Yes, a pooled experimental sample is normally set for normalization and calibration. But in a clinical cohort study, sample collection is commonly on a population level. For MS-based analysis, a large body of samples has to be divided into many batches or analyzed while collecting. So, a pooled sample for calibration is more suitable for ready cohort samples in small sizes.     

Line 295: the number of patients is slightly unbalanced (236 control group vs 1292 cancer group). Have the authors considered this limitation in their statistic evaluations?

The cancer groups mainly consist of 696 lung cancers, 265 gastric cancers, and 279 colorectal cancers. In the case-control analysis, a single cancer type will be compared with the control group. To account for imbalanced group sizes of the control and cancer group, oversampling and undersampling methods will be introduced by using the synthetic minority over-sampling technique (SMOTE) (Chawla et al., 2002, DOI:10.1613/jair.953) and a kNN-based algorithm (Zhang & Mani, 2003, https://www.site.uottawa.ca/~nat/Workshop2003/jzhang.pdf), respectively.

Round 2

Reviewer 1 Report

Missing replies to reviewers comments.

Author Response

Dear reviewer,

many thanks for your comments. We further provided information and explanation for your consideration.

For fasting and cancer pilot study cases, we provided the source data as supplements. Which were also marked in the manuscript (lines 266 and 371). 

For your second comment, if we understood correctly, the explantions are as follows. During data processing, we combined the features across all three platforms and did the deduplication of the metabolites detected by multiple analytical platforms.  Which was conducted on the basis of existing knowledge: GC-MS data were preferred for sugars, amino acids, hydroxy short-chain fatty acids, TCA cycle-involved metabolites, urea, uric acid, creatinine, and creatine were detected by both LC-MS and GC-MS; for long-chain fatty acids with more than 10 carbons, LC-MS lipid data were preferable because they are more soluble in the lipophilic phase. Metabolites with RT between 2 and 14 minutes were preferred for duplication across LC-MS polar and lipid platforms. Despite of above criteria, annotations with better peak shape, higher mean intensity, and higher abundance are favored. For example, the GC-MS data were preferred for alpha-Ketoglutaric acid when it was detected by both LC-MS polar and GC-MS platforms because 1) it is a TCA cycle-involved metabolite; 2) its RT in LC-MS polar platform was before 2 minutes; 3) although its abundance in GC-MS is lower in GC-MS than in LC-MS polar (70% vs. 80%), its annotation score is higher in GC-MS. Another example was hydrocortisone, LC-MS polar data were preferred when it was detected by both LC-MS polar and LC-MS lipid platforms, with higher mean intensity, preferable RT (2.8 vs. 0.7 minutes), higher abundance (95% vs. 46%). Thereafter, the cleaned data was used for statistical and biological analysis. The significantly altered metabolites were compared with the well-known biomarkers of each case (line 278-282 and line 525-532 for the fasting case, line 412-416 and line 552-556 for the CRC case). Furthermore, we did biological analysis for the perturbed metabolites (figure 2D, line 282-284 and line 532-539 for fasting case, line 417-418 and figure 6B for the CRC case). 

For data normalization, we used a deep learning model (normAE) and followed the steps as Zhiwei Rong et al. described (https://dx.doi.org/10.1021/acs.analchem.9b05460). During the stage of implementation, hyperparameters adjustment was set up as default. Briefly, a non-linear autoencoder was used to remove batch effects. It embeds original high-dimensional data into low-dimensional space that contains biological information without inter-batch effect. Data reconstruction was performed subsequently to transform those representations in low-dimensional space back into high-dimensional data to remove inter-batch effect.  Furthermore, adversarial regularization and discriminators were used for optimization of autoencoder to identify and remove the intra-batch effect. (2.7 data normalization line 236-241)

Round 3

Reviewer 1 Report

Although the paper was reasonably presented, it has two major weaknesses.

1. The authors wish to present their metabolomic methods used in their research to the public in the open access journal. If so, they need to disclose full metabolite information, as I have requested

If not, as they explain in their response, they must first clarify their business objectives and then they can submit their work for publication. This is consistent with the general publication principles of science and also with the journal's policy.

2. The problem of "deduplication":

Authors want to present the performance of their methods in metabolomics research. Consequently, the situation is reversed. Measurement of the same metabolites with different platforms should show the same trend if the methods presented have the performance reported in the submitted work

"Deduplication" can only be justified in future studies if the authors show in this work that this criterion is met.

Author Response

Dear reviewer,
thanks for your comments.
For your first comment, I apologize for the mistake we made in providing the source data. We already added them in supplementary materials (Source data 1-2), which include the metabolites information of the fasting case and the cancer pilot study.
For your second comment, we took the CRC case as an example and calculated the foldchange and P values of the metabolites which were identified by multiplatforms. As attached, you can find the detail. Out of 1207 annotated metabolites, there were 43 metabolites identified by two or three platforms. Of these, 40 metabolites were changed with the same trends on different platforms. There were three compounds, such as serotonin, diethanolamin and caffeine, that exhibited different change trends on polar and lipid platforms.In dealing with these cases, we comprehensively consider the physicochemical properties and MS performance and then perform the deduplication. We also added the principle of deduplication into Method 2.6. 

Best regards, 
Yan Li

Round 4

Reviewer 1 Report

Authors, in my opinion, have increased the valuable content of the contribution. Nevertheless, the methodological paper requires further improvements:

1. The data set 3 to the CRC case study should be added (920 annotated features).

2. The data sets should be adequately annotated, each revealed metabolite should have  relevant retention time and elemental composition assigned.

3. The SOP, not only important for the sample preparation, but also for the data sample management, and it should also be included (ie. QC, blank samples, standards, sample orders etc.)

4. LC-MS separation conditions are missing (column, gradient, mobile phase).

5.  I did not find  comments to the evaluation and comparison of the used methods. How did the duplicated-triplicated annotated metabolites work ?

There is an experience that using a different platform, some metabolites may exhibit opposite trends. Therefore this feature should be discussed in the work. 

6. How the non-linear encoder was useful for removal of batch effects? Where are the F1 score data commented ?

Author Response

Dear reviewer,

many thanks for your suggestions and comments. We provided further information for your consideration.

  1. The data set 3 to the CRC case study should be added (920 annotated features).

CRC case study is a sub-cohort of the cancer pilot study. Source data 2 covered all the features of the CRC case.

  1. The data sets should be adequately annotated, each revealed metabolite should have relevant retention time and elemental composition assigned.

Please check out Source data 1 and 2. We provided the information of each annotated metabolite, including MS platform, ESI (+/-), RSD_raw, RSD_norm, m/z, compound formula, and compound name. Due to competing commercial interests, we cannot provide the RT of metabolites publicaly. If there are resonable requests from readers, the dataset is available from the corresponding author.

  1. The SOP, not only important for the sample preparation, but also for the data sample management, and it should also be included (ie. QC, blank samples, standards, sample orders etc.)

For this information, we follow the working guidelines of Fiehn lab and Warwick B. Dunn team. Please check out the details from our method part and references [18,19]. 

  1. LC-MS separation conditions are missing (column, gradient, mobile phase).

Please check the supplemental File S3. There are parameters, such as column, column temperature, autosampler temperature, mobile phase A and B, mobile phase gradient separation, mobile phase flow rate, and complete information for Mass spectrometry. More details could be found in File S3. We also mentioned in the last sentence of 2.5 GC-MS and UPLC-MS detection.

  1. I did not find comments to the evaluation and comparison of the used methods. How did the duplicated-triplicated annotated metabolites work?

For the comparison of the used methods, we compared the performance of three methods for data normalization, NormAE, Median and TIC. The result was provided in figure S5 and mentioned in 3.4 data normalization

There is an experience that using a different platform, some metabolites may exhibit opposite trends. Therefore this feature should be discussed in the work. 

We added one paragraph in Discussion 4.2 for these duplicated-triplicated annotated metabolites.

In addition, during data processing, we combined the features across all three platforms and did the deduplication of the metabolites detected by multiple analytical platforms. Which was conducted based on existing knowledge: 1) GC-MS data were preferred for sugars, amino acids, hydroxy short-chain fatty acids, TCA cycle-involved metabolites, urea, uric acid, creatinine, and creatine when these features were detected by both LC-MS and GC-MS; 2) for long-chain fatty acids with more than 10 carbons, LC-MS lipid data were preferable because they are more soluble in the lipophilic phase; 3) metabolites with RT between 2 and 14 minutes were preferred for duplication across LC-MS polar and lipid platforms. Despite of above criteria, annotations with better peak shape, higher mean intensity, and higher abundance are favored. For example, the GC-MS data were preferred for α-ketoglutaric acid when it was detected by both LC-MS polar and GC-MS platforms because 1) it is a TCA cycle-involved metabolite; 2) its RT in LC-MS polar platform was before 2 minutes; 3) although its abundance in GC-MS is lower than in LC-MS polar (70% vs. 80%), its annotation score is higher. Another example was hydrocortisone, LC-MS polar data were preferred when it was detected by both LC-MS polar and LC-MS lipid platforms, with higher mean intensity, preferable RT (2.8 vs. 0.7 minutes), and higher abundance (95% vs. 46%). Furthermore, some metabolites exhibited different change trends on different platforms. For example, there were 3 out of 1109 metabolites of CRC case study were changed not consistently on polar and lipid platforms. In addition to the above criteria and methods, we selected the platform data by assessing the significance of change on different platforms (i.e., Foldchange and p values). Thereafter, the cleaned data was used for statistical and biological analysis.

  1. 1)How the non-linear encoder was useful for removal of batch effects?

Batch effect was observed as a result of the variation of the instrumental response induced by the maintenance of LC-MS instruments and other systematic effects, which was nonlinear and hardly separate from biological variation. By far, two main methods have been used to reduce the batch effects, which are location-scale methods and matrix factorization methods. In matrix factorization methods, components that correlate to the batch labels identified and the variation derived from identified components would be removed. At this point, nonlinear feature transformations were induced to generate the autoencoder model.

  1. 2)Where are the F1 score data commented?

The metrics of F1 score was calculated as: F1 Score = 2 * (Precision * Recall) / (Precision + Recall) (already added in the manuscript, reference [40]). F1 score is a weighted harmonic mean of precision and recall and was used as the metric to evaluate our performance of machine learning model, which indicates a better performance when it close to 1 and a worse performance when close to 0. The data was visualized in Figure 7 and reported in the correlated part (3.5).  

Best regards,

Yan Li